# Oxytocin neurons drive melanocortin circuit maturation via vesicle release during a neonatal critical period

Pierre-Yves Barelle[1,2], Fabienne Schaller[3], Soyoung Park[4], Emilie Caron[1,2], Jessica Klucznik[1,2], Phillipe Ciofi[5], Françoise Muscatelli[3], Sebastien G. Bouret[1,2]*

1 University of Lille, Inserm, CHU Lille, Laboratory Lille Neuroendocrinology, Lille Neuroscience and Cognition, Lille, France, 2 FHU 1000 Days for Health, School of Medicine, Lille, France, 3 Institut de Neurobiologie de la Méditerranée (INMED), INSERM, Aix Marseille Université, Marseille, France, 4 Children's Hospital Los Angeles, Los Angeles, California, United States of America, 5 Université de Bordeaux, Inserm, Neurocentre Magendie, Bordeaux, France

* sebastien.bouret@inserm.fr

## Abstract

The hypothalamus is crucial for regulating essential bodily functions, including energy balance. It is an exceedingly complex and heterogeneous brain region that contains a variety of neuronal systems that are interconnected with each other. Among these, the melanocortin system, which comprises pro-opiomelanocortin (POMC) and agouti-related peptide (AgRP) neurons, displays a remarkable anatomical relationship with oxytocin (OT) neurons in the paraventricular nucleus (PVH). Here, we demonstrate that OT neurons are instrumental in the development of the melanocortin system in mice. Chemogenetic inhibition of OT neurons during the first postnatal week selectively disrupts POMC and AgRP projections to the PVH, without affecting other target nuclei like the dorsomedial nucleus. This developmental role is age-dependent, as silencing OT neurons in juvenile or adult stages has no impact on melanocortin circuits. OT neurons release various neuropeptides and neurotransmitters, and their secretion can be modulated by chemogenetic manipulation. Expressing the botulinum toxin serotype B light chain in OT neurons reveals that their developmental actions rely on SNARE-mediated exocytosis. Moreover, administering an OT receptor antagonist during the first postnatal week leads to similar melanocortin circuit defects and long-term metabolic effects. Furthermore, neonatal chemogenetic activation of OT neurons rescues POMC circuit deficits in a mouse model of Prader–Willi Syndrome. These findings reveal that OT acts as a paracrine neurotrophic factor orchestrating the development of melanocortin circuits during a restricted neonatal critical period.

**Data availability statement:** All relevant data are within the paper and its Supporting information files.

**Funding:** This work was supported by the Foundation for Prader–Willi Research (grant #697059 to SGB and FM, https://www.fpwr.org), Agence National pour la Recherche (grant ANR-22-CE16-0007, MATBIOTA to SGB, https://anr.fr), Fondation pour la Recherche sur le Cerveau (to SGB, https://www.frcneurodon.org), Horizon2220 program (grant number GA 101080219, eprObes to SGB, https://www.horizon-europe.gouv.fr). The funders had no role in study design, data collection and analysis, decision to publish, or preparation of the manuscript.

**Competing interests:** I have read the journal's policy and the authors of this manuscript have the following competing interests: SGB is a member of PLOS Biology's Editorial Board. The other authors declare that no competing interests exist.

**Abbreviations:** α-MSH, alpha-melanocyte-stimulating hormone; AgRP, agouti-related peptide; ARH, arcuate nucleus of the hypothalamus; Cmpd-21, compound 21; DMH, dorsomedial nucleus of the hypothalamus; EE, energy expenditure; FISH, fluorescent in situ hybridization; LHA, lateral hypothalamic area; nNOS, neuronal nitric oxide synthase; OT, oxytocin; OTR, oxytocin receptor; OTRA, oxytocin receptor antagonist; POMC, pro-opiomelanocortin; PVH, paraventricular nucleus of the hypothalamus; PWS, Prader–Willi Syndrome; ROI, region of interest; VMH, ventromedial nucleus of the hypothalamus.

## Introduction

The hypothalamus is an exceedingly complex brain region that is critical to vital functions. It contains a variety of neuronal systems that must be organized in defined circuits to ensure body homeostasis in response to changes in internal and environmental conditions. The hypothalamic melanocortin system plays a particularly important role in regulating energy balance and adiposity. It is composed of neurons that produce alpha-melanocyte-stimulating hormone (α-MSH), a peptide derived from pro-opiomelanocortin (POMC), and neurons that contain agouti-related peptide (AgRP), an endogenous inverse agonist at α-MSH receptors. POMC neurons exhibit anorexigenic effects, whereas AgRP neurons display orexigenic effects. Both neuronal populations are predominantly located in the arcuate nucleus of the hypothalamus (ARH) and send axonal projections to other hypothalamic nuclei, including the paraventricular nucleus (PVH) [1]. The PVH is strategically located to integrate hormonal and neural information. It sends axonal outputs to various brain structures and to the posterior pituitary to control behaviors and bodily functions. Distinct PVH cell types integrate melanocortin's signal to regulate feeding behavior and energy balance, including neurons producing oxytocin (OT) [2]. Initially identified as a hormone influencing parturition and lactation, OT is now recognized for its broader pleiotropic effects, influencing behavioral and physiological functions ranging from social interaction to stress response and metabolic regulation [3].

Optimal hypothalamic function depends on the establishment of its complex neuronal connectivity during development. In rodents, POMC and AgRP neuronal projections develop during the first two postnatal weeks [4]. The development of OT neurons precedes that of melanocortin circuits [5], suggesting that it could play a key role in the development and functional integration of POMC and AgRP axonal projections. This hypothesis is supported by evidence showing OT's involvement in brain maturation. For instance, OT triggers an excitatory-to-inhibitory switch of GABA actions in the fetal hippocampus shortly before birth [6]. Moreover, OT influences the formation and refinement of neural circuits within the amygdala with long-term effects on emotional regulation and social behavior [7]. In addition, evidence suggests that OT may exert enduring neurobehavioral effects when administered during early life. For example, a single administration of OT at birth corrects suckling deficits in the *Magel2* KO mouse model of Prader–Willi Syndrome (PWS) [8]. Also, neonatal OT treatment in the same mouse model causes long-term beneficial effects on social behavior and learning, whereas adult treatment has only a transient effect [9].

In the present paper, we investigated the importance of OT neurons in the development of melanocortin circuits. Our findings demonstrate that temporarily silencing OT-producing neurons during the first week after birth leads to enduring alterations in the connectivity between both POMC and AgRP neurons and the PVH. We further establish that the developmental influence of OT neurons on melanocortin circuits is facilitated through SNARE-mediated exocytosis and necessitates the binding of OT to its receptor. Additionally, our research reveals that chemogenetically stimulating OT neurons can rescue the abnormal development of POMC circuits in a mouse model

of PWS. This research underscores the significance of OT signaling during early life stages in shaping neural networks essential for metabolic regulation.

## Results

### Close anatomical association between POMC and AgRP fibers and oxytocin neurons in the paraventricular nucleus

ARH projections are largely intrahypothalamic, with a great density of fibers and terminals found in the PVH [4]. We used a three-dimensional proximity analysis of POMC- and AgRP-immunoreactive fibers to investigate their spatial relationships with OT neurons in the PVH. We found that POMC and AgRP axons converge preferentially on OT cell bodies and processes (Fig 1A and 1C). Half of POMC- and AgRP-immunoreactive fibers found in the PVH are located less than 5 µm of OT-positive neurons with up to 25% of fibers detected at less than 2 µm of OT cells (Fig 1B and 1D). Strikingly, the presence of POMC and AgRP fibers was mainly observed in the ventral portion of the PVH, where OT neurons are predominantly located, leaving a central zone within the PVH partially devoid of fibers (Fig 1A). This distinctive arrangement of POMC and AgRP fibers towards OT neurons suggests that OT may act as a neurotrophic factor to direct POMC and AgRP fibers toward their direction.

### Neonatal chemogenetic inhibition of oxytocin neurons disrupts melanocortin circuits

To examine whether OT-containing neurons influence the development of melanocortin circuits, we first developed a transgenic mouse model in which OT neurons are chemogenetically inhibited by crossing *Ot*-Cre mice with

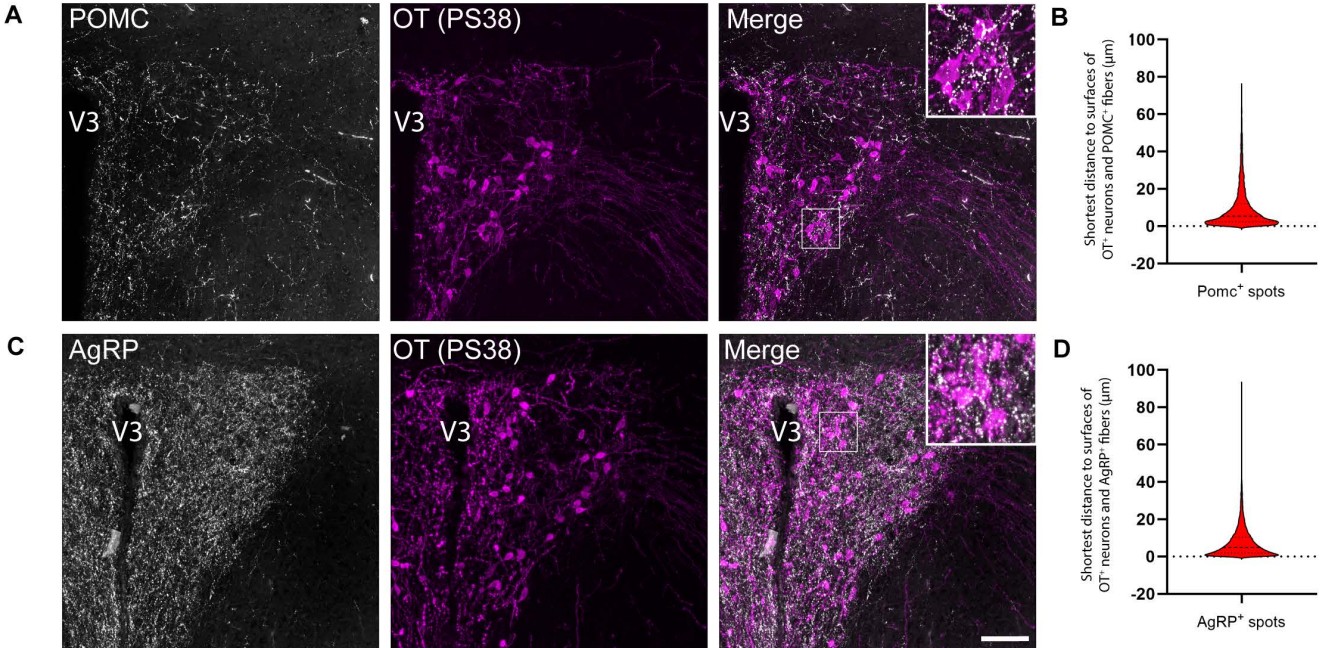

**Fig 1. Close anatomical association between POMC and AgRP fibers and oxytocin neurons in the paraventricular nucleus. (A, C)**. Representative images of OT (PS38) immunoreactive neurons and **(A)** POMC- and **(C)** AgRP-immunoreactive fibers in the paraventricular nucleus of the hypothalamus (PVH) of adult wild-type mice. **(B, D)**. Quantification of the shortest distance between **(B)** POMC- or **(D)** AgRP-immunoreactive fibers and OT-immunoreactive cell bodies processes within the PVH (*n* = 5 animals per group). V3, third ventricle. Scale bar, 50 µm. The data underlying this Figure can be found in S1 Data.

Rosa-CAG-LSL-HA-hM4Di-p2a-mCitrine mice (Fig 2A). We confirmed that the inhibitory DREADD was selectively expressed in OT neurons by observing mCitrine fluorescence, which was included in the targeting vector, in OT-immunopositive neurons (Fig 2B). To determine whether OT-containing neurons exert neurodevelopmental actions during critical period(s), we chemogenetically silenced OT neurons during different periods of postnatal life by injecting the DRE-ADD ligand called compound 21 (Cmpd-21) during the following developmental periods: neonatal (P0–P7), juvenile (P25–P32), and adulthood (P53–P60). The advantage of using Cmpd-21, in comparison to clozapine-N-oxide injections, is that Cmpd-21 is a very potent full agonist of hM4Di, it has a superior brain penetration and long-lasting presence, and does not cause off-target effects on a wide range of behavioral measures [10]. Whole-cell current-clamp recordings confirmed that Cmpd-21 inhibited action potential firing in OT neurons from *Ot*-Cre::R26-LSL-hM4Di-DREADD mice [11].

Chemogenetic inhibition of OT-containing neurons during the neonatal life in males resulted at adulthood in a 2.4- and 1.8-fold reduction in the density of POMC- and AgRP-immunoreactive fibers innervating the PVH, respectively (Fig 2C and 2D). Interestingly, this effect appeared specific to the PVH since POMC and AgRP projections to the dorsomedial nucleus of the hypothalamus (DMH) were unaffected (Fig 2E and 2F). In contrast, chemoinhibition of OT-containing neurons during the juvenile period or the adult life had no impact on the density of POMC- or AgRP-containing projections to either the PVH or the DMH (Fig 2C–2F). We also checked whether chemoinhibition of OT-containing neurons could impact OT neurons themselves and found no alterations in the oxytocinergic system, as evidenced by a normal number of OT cells in the PVH (Fig 2G) and an unaltered density of OT-immunoreactive fibers in the lateral hypothalamic area (LHA) (Fig 2H). Notably, we found that the effects of OT chemoinhibition were sex-specific, with females not being affected (S1 Fig).

## Normal development of melanocortin circuits in mice lacking nitric oxide

Previous studies have demonstrated that a subset of PVH neurons express neuronal nitric oxide synthase (nNOS) [12]. We investigated whether these nNOS-positive cells found in the PVH could be oxytocinergic neurons and found that the vast majority of OT-positive cells in the PVH co-express nNOS during neonatal life (Fig 3A). Because nitric oxide (NO) has been implicated in axonal growth and guidance [13,14], and because activation of hM4Di has been shown to alter NO production [15], we investigated whether the aberrant POMC and AgRP circuits seen after chemogenetic inhibition of OT neurons could be mediated through NO by examining melanocortin circuits in *nNOS* KO mice. The results showed that there were no alterations in the density of POMC and AgRP neuronal projections to the PVH (Figs 3B, 3C and S1) and DMH (Figs 3D, 3E and S1) in male and female *nNOS* KO mice. In addition, the oxytocinergic system appears normal in mutant mice (Figs 3F, 3G and S1). These results indicate that whereas many OT neurons co-express NO, this gaseous neurotransmitter is not involved in the establishment of melanocortin circuits.

## Oxytocin neurons influence the development of melanocortin circuits through a SNARE protein-dependent process that requires functional vesicle-associated proteins

The results described above suggest that the neurodevelopment effects of OT-producing neurons' chemoinhibition are not mediated by diffusible molecules such as NO, but rather involve the release of large molecules such as neuropeptides and neurotransmitters. To test this hypothesis, we used a conditional mouse model expressing the botulinum toxin sero-type B light chain (BoNT/B), causing inhibition of SNARE-mediated exocytosis [16]. These mice were crossed with *Ot*-Cre mice to induce OT neuron-specific blockade of SNARE-dependent exocytosis (Fig 4A).

Blockade of vesicular fusion in OT-containing neurons of male, but not female, mice resulted in a 1.8-fold reduction in the density of POMC- and AgRP-immunoreactive fibers innervating the PVH (Figs 4B, 4C and S1). Similar to the chemo-inhibition of OT neurons, the effect of exocytosis blockade on melanocortin circuits was specific to the innervation of the PVH, as the density of POMC and AgRP projections to the DMH was comparable between *Ot*-Cre::BoN/T and control

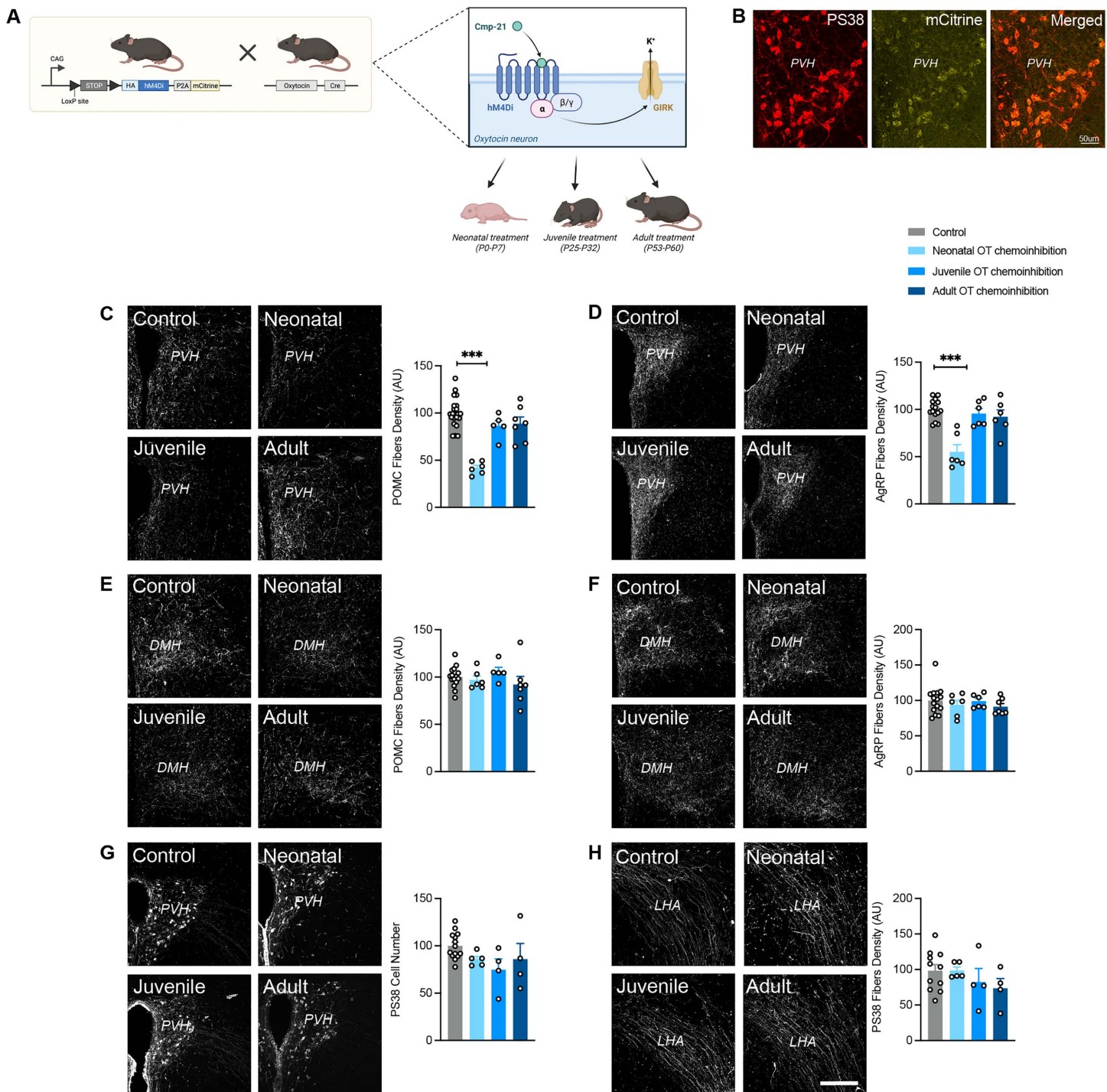

**Fig 2. Neonatal chemogenetic inhibition of OT neurons disrupts melanocortin circuits. (A)** Description of the animal model. *Created in BioRender. Bouret, S. (2025)* https://BioRender.com/v6d7iia. **(B)** Representative images of OT neurons containing mCitrine in the paraventricular nucleus (PVH). **(C–F)** Representative images and quantification of **(C, E)** POMC and **(D, F)** AgRP fibers innervating the **(C, D)** PVH and **(E, F)** dorsomedial nucleus (DMH) of adult R26-LSL-hM4Di-DREADD (control) and *Ot*-Cre::R26-LSL-hM4Di-DREADD injected with Compound-21 neonatally, or during juvenile life, or during adulthood (*n* = 6–7 animals per group). **(G, H)** Representative images and quantification of OT (PS38+) neurons and fibers innervating the **(G)** PVH and **(H)** lateral hypothalamic area (LHA), respectively, in adult R26-LSL-hM4Di-DREADD (control) and *Ot*-Cre::R26-LSL-hM4Di-DREADD injected with Compound-21 neonatally, or during juvenile life, or during adulthood (*n* = 4–5 animals per group). Data are presented as mean + SEM. \*\*\**P* < 0.001. Scale bar, 200 μm. The data underlying this Figure can be found in S1 Data.

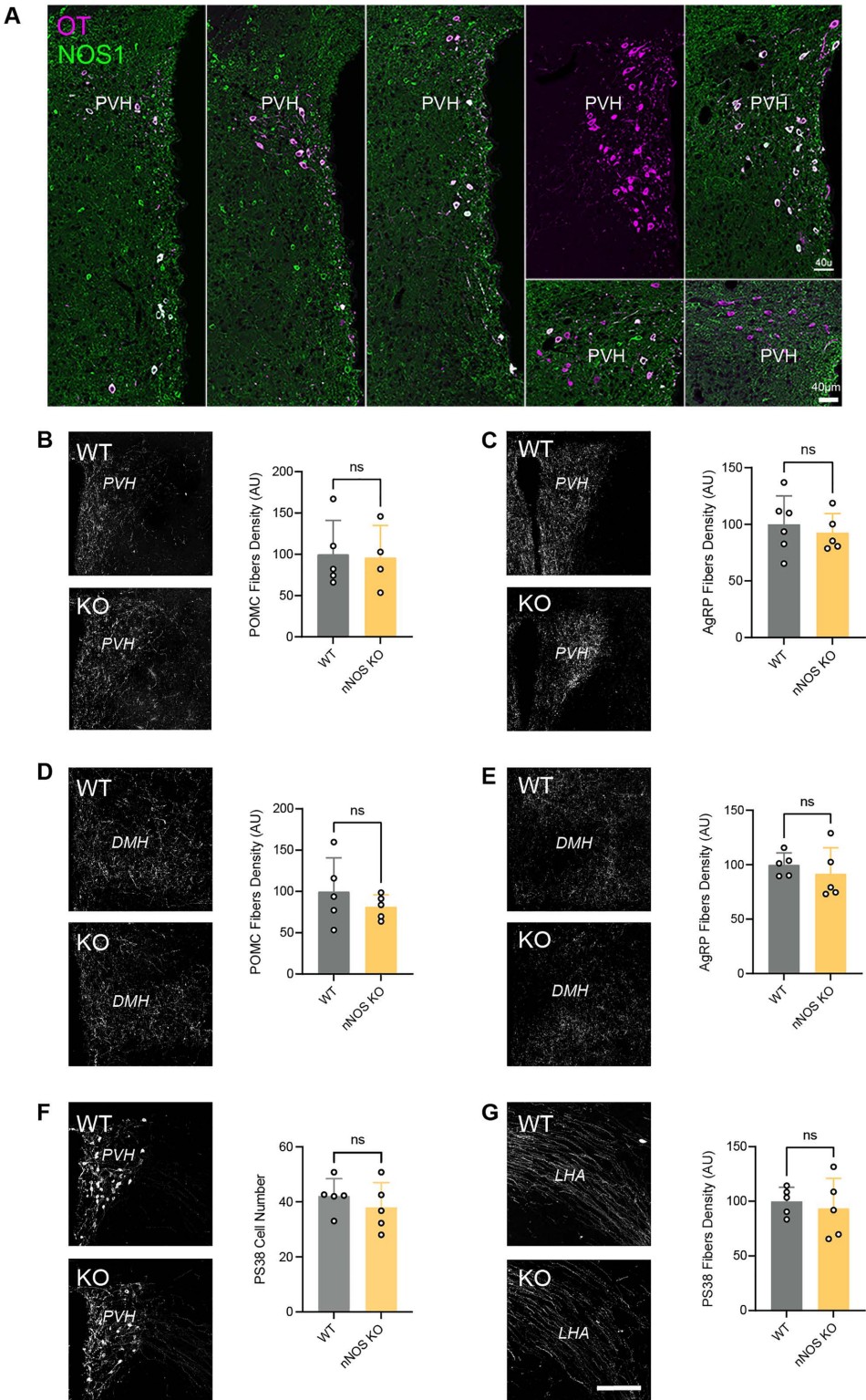

**Fig 3. Normal development of melanocortin circuits in nitric oxide-deficient mice. (A)** Representative images of single optical sections of the hypothalamus of P10 mice showing OT and nitric oxide co-labeling in the paraventricular nucleus (PVH). **(B–E)** Representative images and quantification of **(B, D)** POMC and **(C, E)** AgRP fibers innervating the **(B, C)** PVH and **(D, E)** dorsomedial nucleus (DMH) of adult wild-type (control) and *nNOS* KO mice

(*n* = 4-5 animals per group). **(F, G)** Representative images and quantification of OT (PS38+) neurons and fibers innervating the **(F)** PVH and **(G)** lateral hypothalamic area (LHA), respectively, in adult wild-type (control) and *nNOS* KO mice (*n* = 5 animals per group). Data are presented as mean + SEM. Scale bars, 40 μm **(A)**, 200 μm **(B–G)**. The data underlying this Figure can be found in S1 Data.

mice (Figs 4D, 4E and S1). Similarly, the exocytose blockade did not cause neuroanatomical alterations of the oxytocinergic system (Figs 4F, 4G and S1).

Together, these results suggest that the normal development of melanocortin circuitry to the PVH requires exocytosis from OT-containing neurons.

### Developmental pattern of OT projections

To investigate the developmental timeline of OT neuron projections, *Ot*-Cre::mT/mG reporter mice were used in combination with PS38 immunostaining (Fig 5A). OT neuronal cell bodies were detected in the PVH as early as P0. However, at this stage, their axonal projections to other hypothalamic regions were sparse. By P25, OT fibers extended from the PVH into the LHA, the supraoptic nucleus, and the internal layer of the median eminence, with a similar pattern observed at P60 (Fig 5B). Notably, no OT projections (labeled by either mGFP or PS38) were detected within the ARH at any postnatal stage examined (Fig 5B). Additionally, the density and distribution pattern of OT projections were similar between males and females (S2 Fig).

### OTR signaling is required for normal development of melanocortin circuits

OT neurons express a variety of neuropeptides and neurotransmitters such as OT and glutamate and 10% of these neurons also express vasopressin, all of which can be affected in our *Ot*-Cre::R26-hM4Di-DREADD and *Ot*-Cre::BoN/T mouse models. To further explore whether the neurodevelopment effects of OT-containing neurons on POMC and AgRP circuits could be mediated by OT per se, we first investigated whether POMC and AgRP neurons express the oxytocin receptor (OTR). Quantitative analysis of gene expression using RNAscope reveals that *Otr* mRNA is expressed in various nuclei of the hypothalamus, including in the ARH, ventromedial nucleus (VMH), DMH, and PVH, throughout postnatal life (Fig 5C–5E). Among all hypothalamic nuclei examined, the VMH consistently showed the highest expression of *Otr* across all ages tested (Fig 5E). Within the ARH, *Otr* was found in 60.5% of *Pomc* neurons at P0, increasing to 73% by P60 (Fig 5F and 5G). 28.2% of *Agrp* neurons co-expressed *Otr* mRNA at P0, rising to 42.8% at P60 (Fig 5F and 5G). Notably, the majority of *Pomc*- and *Agrp*-expressing neurons expressed relatively low levels of *Otr* mRNA (Fig 5G). The expression pattern of *Otr* in *Pomc* and *Agrp* neurons displayed no marked sex differences (S2 Fig). Together, these observations suggest that OT can potentially act directly on POMC and AgRP neurons as early as birth and throughout postnatal development.

To determine if OT acts directly on arcuate neurons to influence axon growth, we performed a series of experiments in which ARH explants from neonatal WT mice were cultured *ex vivo* and exposed to OT. After 48h, the density of axons extending from arcuate explants in which OT was added to the culture medium was 1.8-fold higher than that of control explants (Fig 6). Together, these data provide evidence that OT acts directly on arcuate neurons to promote axon growth. To investigate the in vivo relevance of these findings, we examined melanocortin circuits in WT mice that were injected daily from P0 to P7 with L-9,889, a specific oxytocin receptor antagonist (OTRA) (Fig 7A). Adult male mice treated with the OTRA neonatally displayed a 2-fold reduction in the density of POMC-immunoreactive fibers innervating the PVH (Fig 7B), but innervation of the DMH was unchanged (Fig 7D). Interestingly, we did not find alterations in melanocortin or OT fibers in females treated with L-368,899 neonatally, nor any changes in AgRP projections in either male or female mice (Figs 7C, 7E–7G and S1).

These results are consistent with our findings in both *Ot*-Cre::R26-hM4Di-DREADD and *Ot*-Cre::BoN/T mouse models, substantiating the requirement of OT secretion and binding to its receptor to promote the development of POMC projections to the PVH during a neonatal critical period.

   

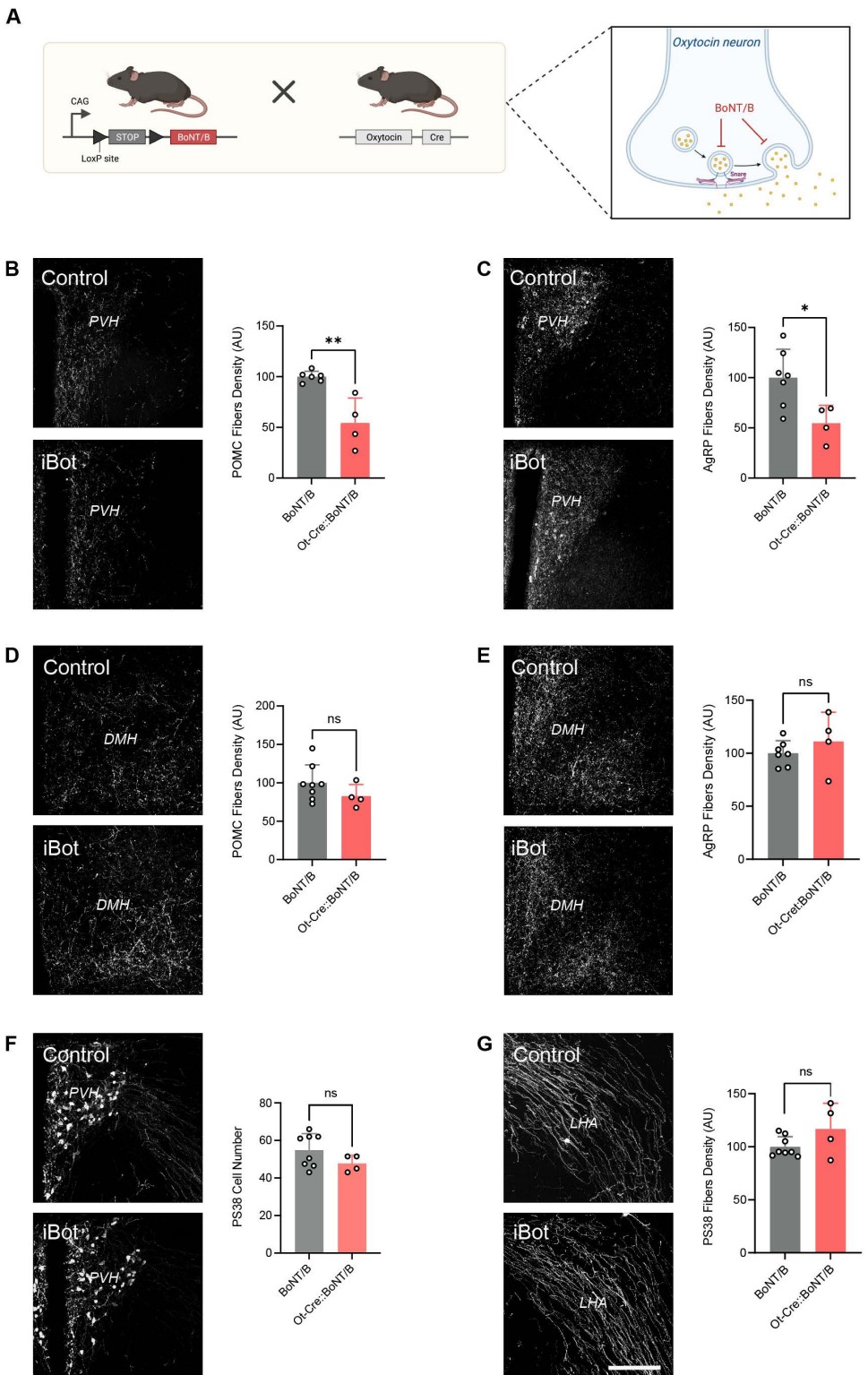

**Fig 4. Blocking exocytosis of OT neurons perturbs the development of melanocortin circuits. (A)** Description of the animal model. *Created in BioRender. Bouret, S. (2025)* https://BioRender.com/v6d7iia. **(B–E)** Representative images and quantification of **(B, D)** POMC and **(C, E)** AgRP fibers innervating the **(B, C)** paraventricular nucleus (PVH) and **(D, E)** dorsomedial nucleus (DMH) of adult *BoNT/B* (control) and *Ot*-Cre::*BoNT/B* mice (*n* = 4−8

animals per group). **(F, G)** Representative images and quantification of OT (PS38+) neurons and fibers innervating the **(F)** PVH and **(G)** lateral hypothalamic area (LHA), respectively, in adult *BoNT/B* (control) and *Ot*-Cre::*BoNT/B* mice (*n* = 8−4 animals per group). Data are presented as mean + SEM. *$P < 0.05$, **$P < 0.01$. Scale bar, 200 μm. The data underlying this Figure can be found in S1 Data.

### Neonatal blockade of oxytocin signaling causes lifelong metabolic dysregulations

We also investigated the long-term physiological consequences of OTR antagonism during neonatal life. Male and female mice treated with the OTRA neonatally had normal body weight (Fig 8A). However, neonatal treatment with the OTRA in male, but not in female, mice resulted in reduced adiposity and elevated lean mass compared to control animals (Figs 8B and S3). It is known that body composition could be a factor influencing glucose regulation. Adult mice treated with L-368,899 neonatally displayed lower fasting glycemia levels than their respective controls (Fig 8K). In addition, neonatal OTR antagonism resulted in improved glucose tolerance after a glucose challenge (Fig 8L and 8M). We also conducted a comprehensive assessment of energy balance regulation of adult mice treated with L-368,899 neonatally. Neonatal OTRA treatment resulted in a significant increase in cumulative and nocturnal food intake (Fig 8C and 8D), and a higher mean and diurnal respiratory exchange ratio (RER) (Fig 8E and 8F), suggesting a shift toward a more carbohydrate-centric metabolism in these animals. However, locomotor activity, energy expenditure (EE), and z-rearing were unaffected in OTRA-treated animals (Fig 8G–8J).

### Neonatal chemogenetic activation of oxytocin neurons rescues POMC circuit deficits in *Magel2* KO mice

Dysregulation of the OT system has been linked to various forms of obesity, including of genetic origins such as PWS [17]. Interestingly, the *Magel2* KO mouse model of PWS displays a marked reduction in OT content at birth [8] associated with alterations in POMC neurocircuitry [18]. Based on these observations, we used a stimulatory Cre-dependent DREADD approach to explore the ability of OT neurons to functionally rescue the neurodevelopmental deficits of POMC circuits in *Magel2* KO mice. To do so, we crossed *Ot*-Cre::R26-hM3Dq-DREADD mice with *Magel2* KO animals (Fig 9A). We specifically chemogenetically stimulated OT neurons in *Magel2* KO mice during the first week of postnatal life, given our findings described above revealing a critical neonatal period for the neurodevelopmental actions of OT on melanocortin circuits. We first validated our animal model by injecting Cmpd-21 in *Ot*-Cre::R26-hM3Dq-DREADD mice and observing specific activation of OT neurons using cFos immunostaining (Fig 9A). Our results revealed a significant increase in the density of POMC-immunoreactive fibers in the PVH of *Magel2* KO, *Ot*-Cre::R26-hM3Dq-DREADD mice treated with Cmpd-21 neonatally (Fig 9B). The chemoactivation of OT neurons was specific to the POMC innervation of the PVH, since the density of POMC projections to the DMH, and the density of AgRP fibers to the PVH appeared comparable between *Magel2* KO, R26-hM3Dq-DREADD and *Magel2* KO, *Ot*-Cre::R26-hM3Dq-DREADD mice (Fig 9C and 9D). Surprisingly, the density of AgRP fibers was increased in the DMH of *Magel2* KO, *Ot*-Cre::R26-hM3Dq-DREADD mice treated with Cmpd-21 neonatally (Fig 9E). However, the chemogenetic activation of OT neurons did not affect the OT fibers in *Magel2* KO mice (Fig 9F and 9G).

### Discussion

POMC and AgRP neurons send extensive axonal projections to the PVH, particularly toward OT neurons, to modulate feeding and metabolism. However, whether OT neurons actively contribute to the development of melanocortin circuitry is unknown. The present study shows that chemogenetic inhibition of OT-expressing neurons during the neonatal period, but not during the juvenile or the adult period, disrupts POMC and AgRP innervation of the PVH. We also sought to elucidate the factors and mechanisms underlying these neurodevelopmental effects since OT neurons can produce a variety of neuropeptides and neurotransmitters including OT, glutamate, vasopressin, but also diffusible factors such as NO, which has been shown to play a role in axonal guidance [13,14,19,20]. Therefore, we investigated whether the neurodevelopmental

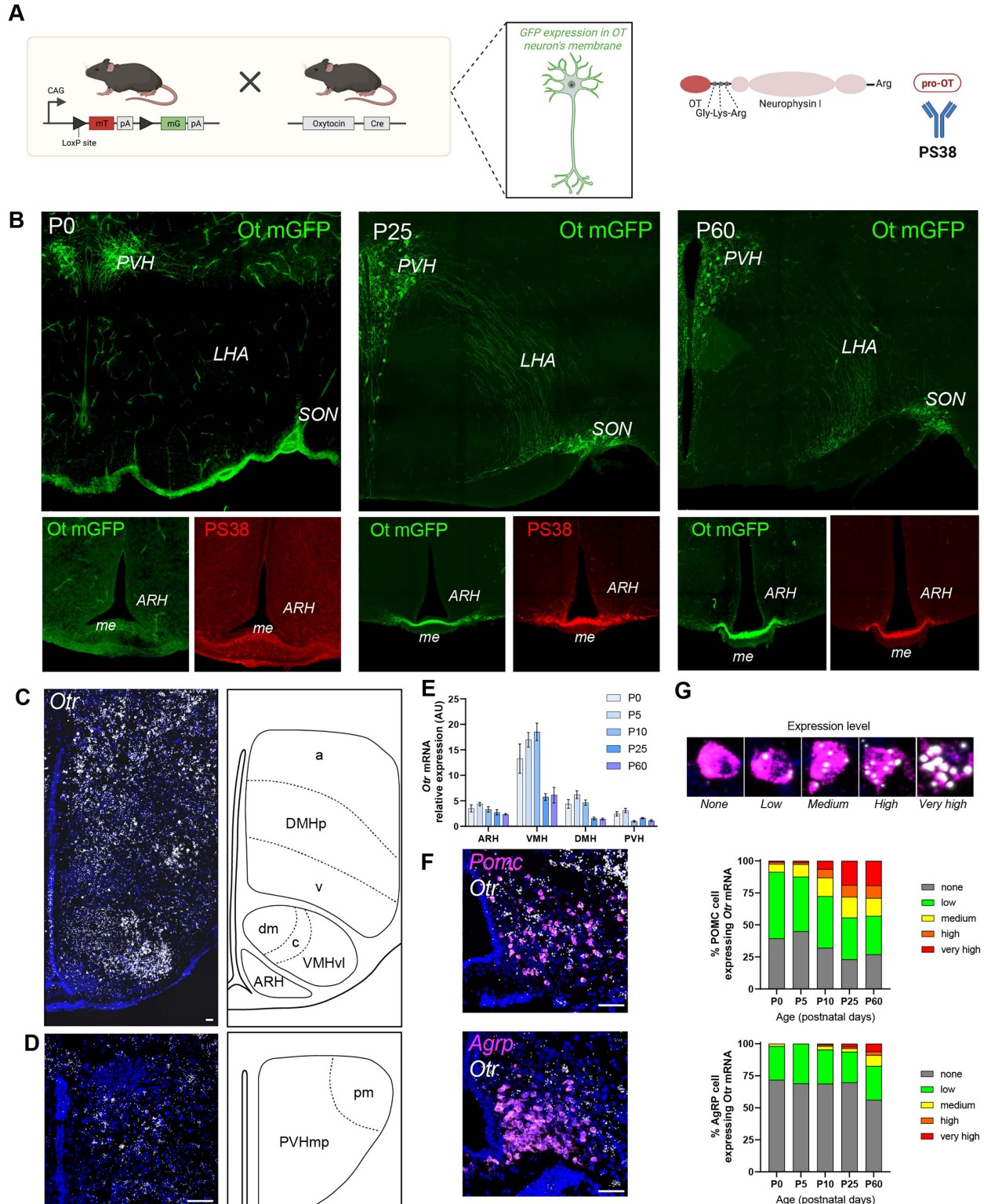

**Fig 5. Distribution of oxytocin fibers and expression pattern of oxytocin receptor in the developing hypothalamus. (A)** Description of the animal model and **(B)** representative images showing immuno- (PS 38 antibody) and genetic (Ot mGFP) labeling of OT neurons and fibers at the levels of the paraventricular nucleus (PVH) and arcuate nucleus (ARH) at P0, P25, and P60. **(C, D)** Representative images showing *oxytocin receptor* (*Otr*)

mRNA at the level of **(C)** the arcuate/ventromedial/dorsomedial nuclei and **(D)** the paraventricular nucleus of a P10 WT mouse. **(E)** Relative expression of *Otr* mRNA in the ARH, VMH, DMH, and PVH of WT mice at P0, P5, P10, P25, and P60 (*n* = 3–8 animals per group). **(F)** Representative images and **(G)** quantification of *Otr co-expression in Pomc and Agrp neurons* at P0, P5, P10, P25, and P60 (*n* = 6–8 animals per group). Data are presented as means ± SEM. LHA, lateral hypothalamic area; me, median eminence; SON, supraoptic nucleus. Scale bar 50 μm. The data underlying this Figure can be found in <u>S1 Data</u>. <u>Fig 5A</u> is *created in BioRender. Bouret,* S. *(2025)* <u>https://BioRender.com/d5iyc5j</u>.

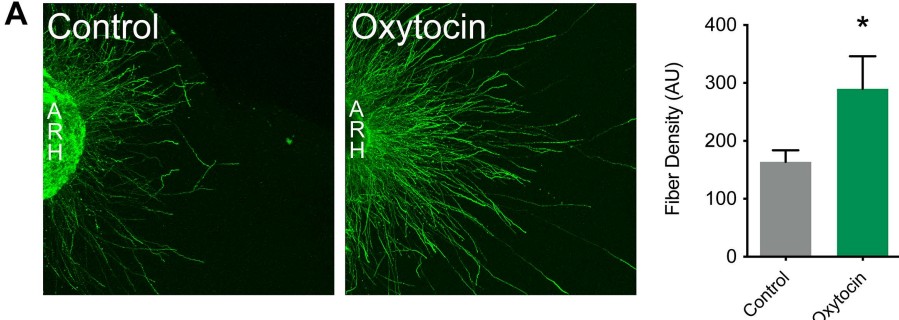

**Fig 6. Oxytocin promotes the growth of arcuate nucleus axons.** Representative images and quantification of βIII-tubulin–immunopositive fibers from isolated cultures of neonatal ARH incubated for 48 hours with vehicle or oxytocin (100 ng/ml) (*n* = 4–8 per group). Data are presented as means + SEM. The data underlying this Figure can be found in <u>S1 Data</u>.

alterations observed upon chemogenetic silencing of OT neurons were triggered by neuropeptides/neurotransmitters secreted by exocytosis from OT neurons or mediated through NO production. No alterations in POMC or AgRP neuronal projections were found in *nNOS* KO mice. However, the development of POMC and AgRP circuits to the PVH was altered in animals expressing botulinum toxin serotype B light chain genes in OT-producing neurons, suggesting that melanocortin circuits' development is dependent on SNARE-mediated exocytosis.

In agreement with existing literature showing a direct role for OT in neuronal development [6,21,22], our study support the hypothesis that OT neurons influence the development of melanocortin circuits. First, the chemogenetic inhibition of OT neurons during neonatal life disrupts POMC and AgRP neural projections. Second, our *ex vivo* arcuate nucleus explant experiments show that direct exposure to OT promotes axonal growth. Third, neonatal administration of an OTR antagonist disrupts the development of POMC projections in vivo. Surprisingly, OTR antagonism did not affect the density of AgRP fibers. Nevertheless, this is consistent with our RNAScope analysis showing that although 60% neonatal POMC neurons express *Otr*, only 30% or AgRP neurons express this receptor. The mechanism by which OT neurons influence AgRP projections remains to be elucidated. However, our findings using *Ot*-Cre::BoNT mice suggest that this may involve the co-release of other neurotransmitters or neuropeptides from OT neurons, such as glutamate and to a lesser extent vasopressin, both of which have been shown to influence axon growth in other neuronal systems [23–26]. The site(s) of action where OT exerts its neurodevelopmental effect on POMC neurons remains to be determined. OT could act directly on POMC neurons at the level of the ARH. However, our findings and others [27] indicate that the ARH is virtually devoid of OT fibers throughout postnatal development. These observations suggest that OT neurons may contribute primarily to the early formation or targeting of POMC and AgRP projections through the exocytosis of axon guidance cues at the level of the PVH. In support of this idea, we have previously shown that PVH neurons secrete class 3 semaphorins, which play a critical role in the development of POMC circuits [28]. Interestingly, a search on the HypoMap single-cell RNA sequencing database (https://cellxgene.cziscience.com/e/dbb4e1ed-d820-4e83-981f-88ef7eb55a35.cxg/) revealed that Sema3A, Sema3C, and Sema3E are enriched in OT neurons. These findings support the idea that OT neurons could influence melanocortin circuit development through semaphorin signaling as well. Alternatively, OT could also be released into the third ventricle [29], thereby reaching ARH neurons, since it has been shown that molecules can freely diffuse between the

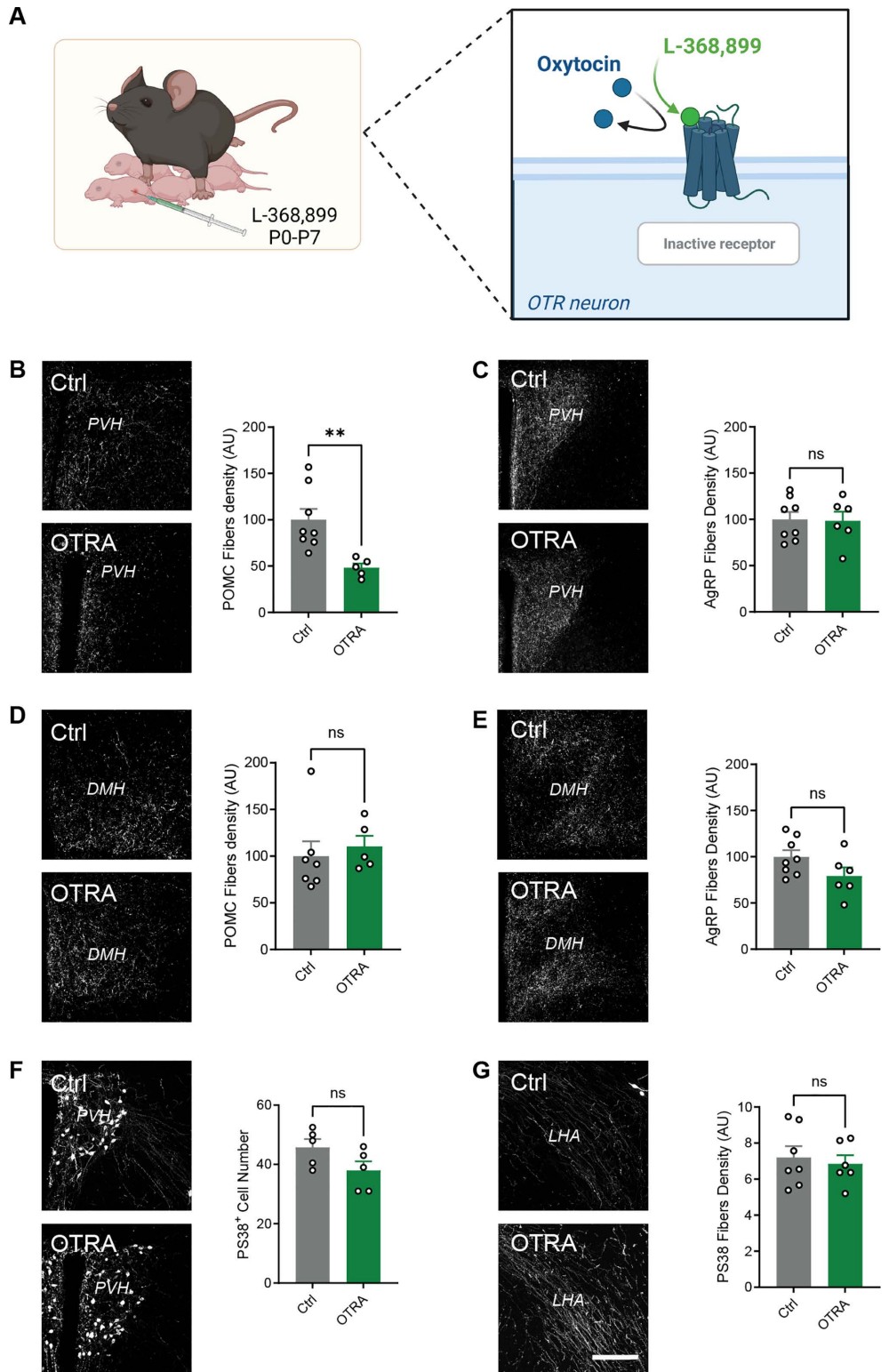

**Fig 7. Neonatal blockade of oxytocin signaling disrupts melanocortin circuits. (A)** Description of the animal model. *Created in BioRender. Bouret, S. (2025)* https://BioRender.com/d5iyc5j. **(B–E)** Representative images and quantification of **(B, D)** POMC and **(C, E)** AgRP fibers innervating the **(B, C)** paraventricular nucleus (PVH) and **(D, E)** dorsomedial nucleus (DMH) of adult wild-type mice injected with saline (control) or an oxytocin receptor

antagonist (L-368,899, OTRA) neonatally ($n=8-5$ animals per group). **(F, G)** Representative images and quantification of OT (PS38+) neurons and fibers innervating the **(F)** PVH and **(G)** lateral hypothalamic area (LHA), respectively, in adult mice injected with saline (control) or an oxytocin receptor antagonist (L-368,899, OTRA) neonatally ($n=7-6$ animals per group). Data are presented as means + SEM. **$P < 0.01$. Scale bar 200 μm. The data underlying this Figure can be found in S1 Data.

cerebrospinal fluid and the ARH [30]. One potential limitation of our study is that it does not definitively demonstrate that OT release per se acts via OT receptors to promote axonal development of POMC and AgRP neurons. Future analyses using loss-of-function models such as inducible POMC- and AgRP-specific OTR knockouts will be required to address this question.

Another notable finding of our study is the marked sex differences observed in the effects of OT neurons on the development of the melanocortin system. A possible explanation could be related to the expression of *Otr* in POMC/AgRP neurons that may be sexually dimorphic. However, our results indicated that the density of *Otr* mRNA expression in POMC and AgRP neurons was relatively similar between males and females. Additionally, there could be inherent differences in OT projections between males and females. However, our data indicate that there are no sex differences in the density of OT terminals in the hypothalamus, including in the arcuate nucleus. Therefore, the marked sexual dimorphism in the effect of OT on the development of the melanocortin system remains unexplained but could involve other mechanisms or factors, such as sex hormones, including estrogens that have been shown to have protective effects on metabolic regulations [31].

The unique metabolic profile of animals treated with the OTR antagonist neonatally remains particularly intriguing. Neonatal OTR antagonism causes an increase in food intake and the use of carbohydrates as a fuel source, but it is also associated with reduced fat mass, increased lean mass, and improved glucose tolerance. This complex phenotype may be explained by the ubiquitous expression of *Otr* throughout the CNS and the body that may be targeted with the OTR antagonism [22,32–36]. For example, the hyperphagia may be the result of altered OTR signaling in POMC neurons since these neurons are known for their anorexigenic action, and we found that *Otr* is expressed in the majority of POMC neurons. Moreover, previous studies have reported that intra-ARH OT injections reduce food intake [37]. However, OT can also act on other neural networks to control food intake. Consistent with this idea, our *ex vivo* explant assay indicates that OT promotes the overall neurite outgrowth of arcuate neurons. Since previous single-cell RNA sequencing studies [38] have revealed that the arcuate nucleus contains up to 34 distinct neuronal populations, it is therefore likely that OT exerts a stimulatory effect on axon growth from a broader range of arcuate neurons, including POMC neurons, but also other non-identified neuronal subtypes that could be involved in the regulation of energy homeostasis. In addition, our RNAscope analysis shows that *Otr* is widely expressed throughout the neonatal hypothalamus, with strong expression in the VMH and DMH, further highlighting potential additional sites of OTRA action during development. In addition, it is possible that OT acts on extra-hypothalamic regions, such as the mesolimbic pathway (*i.e.*, the nucleus accumbens and ventral tegmental area) and particularly its interactions with the dopaminergic system to modulate responses to external social stimuli [39]. There is also the possibility that the OTR antagonist may have influenced peripheral systems involved in glucose and energy homeostasis. For example, *Otr* is highly expressed in adipose tissue [34], suggesting a possible direct effect of OTRA on adipocyte development and function, especially since OTRA was injected during critical periods of adipocyte tissue development [40]. Although the site(s) of action of the OTR antagonist remains to be identified, our findings are intriguing as they show that blocking OT signaling during a discrete period of life is sufficient to cause lifelong metabolic effects, highlighting the importance of controlling optimal OT function during early life.

The pathological relevance of our findings is demonstrated by experiments in *Magel2* KO mice, a mouse model of PWS. *Magel2* KO mice display a marked reduction in OT content at birth and during infancy [8]. Interestingly, the *Magel* KO mouse model also has alterations in POMC neurocircuits [18]. Here, we found that chemogenetic reactivation of OT neurons during a neonatal critical period can restore the defective POMC projections in *Magel2* KO

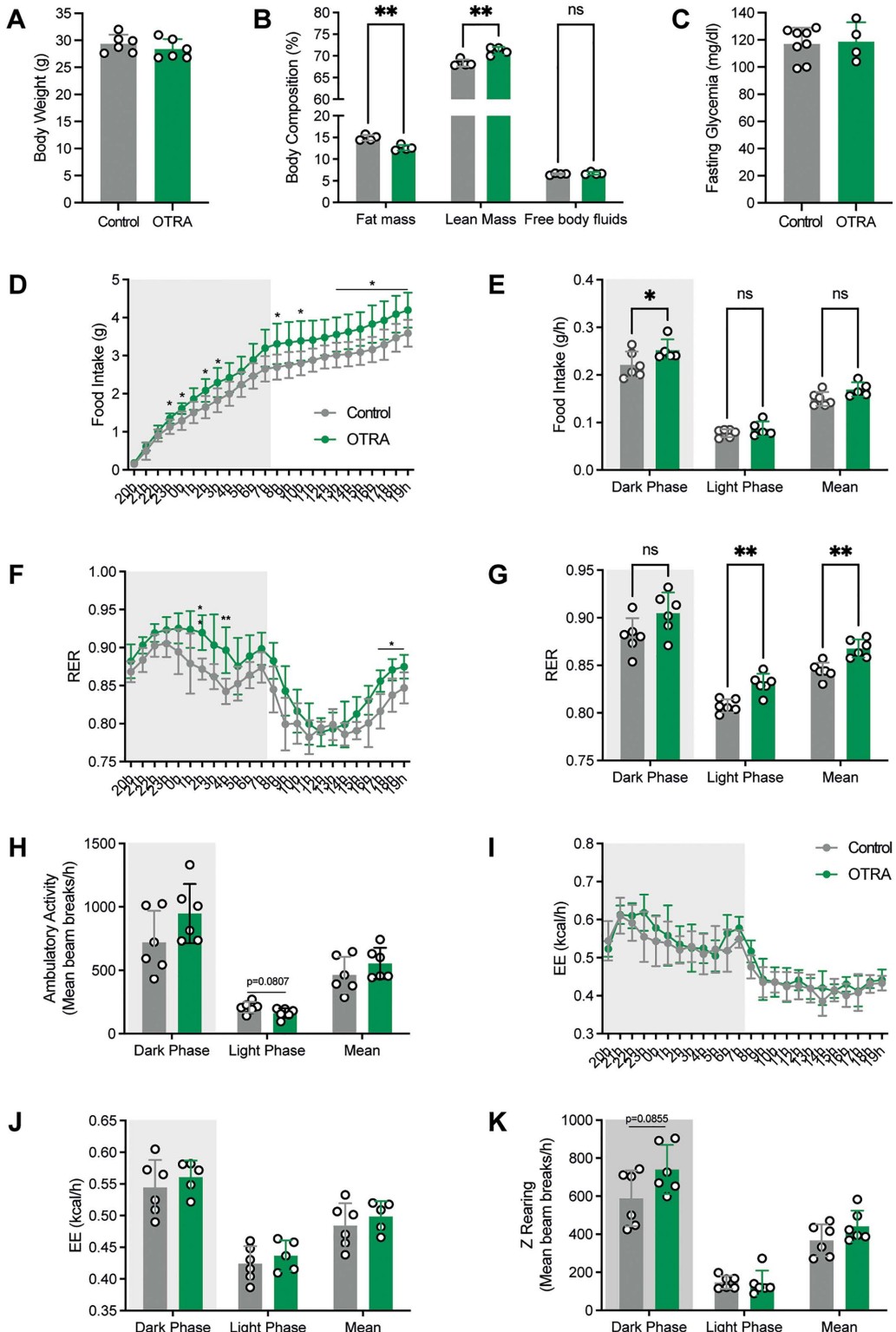

**Fig 8. Neonatal blockade of oxytocin signaling causes lifelong metabolic dysregulations. (A)** Body weight, **(B)** body composition, **(C)** cumulative and **(D)** average food intake, **(E)** 24 h and **(F)** average respiratory exchange ratio, **(G)** average ambulatory activity, **(H)** 24 h and **(I)** average energy expenditure, **(J)** average Z rearing, **(K)** fasting glycemia, **(L)** glucose tolerance test (GTT) and **(M)** area under the GTT curve in adult male mice injected

with saline (control) or an oxytocin receptor antagonist (L-368,899, OTRA) neonatally ($n=6$ animals per group). Data are presented as means±SEM. *$P<0.05$, **$P<0.01$. The data underlying this Figure can be found in S1 Data.

mice. Interestingly, the loss of OT neurons' activity is a hallmark of other perinatally-acquired health conditions such as those linked to maternal obesity [41]. Pups born to obese dams have fewer OT immunoreactive neurons in the hypothalamus, and correcting OT levels using a microbiota intervention rescues social deficits in this offspring [41]. Maternal obesity also causes lifelong weight gain and glucose intolerance associated with a disruption in POMC and AgRP axonal projections during adulthood [42–44]. Whether reactivation of OT functionally rescues abnormal development of the melanocortin system and the associated metabolic defects in this pathological condition remains to be determined. Equally interesting would be to examine whether chemogenetic activation of OT neurons in WT mice enhances POMC projections.

In conclusion, this comprehensive study highlights a novel role for OT neurons in the development of the melanocortin system, with a paracrine neurotrophic effect in the PVH during a neonatal critical period. We have also identified some of the complex neurobiological mechanisms by which OT neurons influence the wiring of melanocortin circuits, highlighting the role of OT exocytosis and OTR-dependent signaling. Additionally, altered metabolic profiles in models with disrupted OT signaling reveal OT's broad influence on metabolic processes, emphasizing the potential long-term impacts of early postnatal manipulations of OT pathways on metabolic health. This research not only highlights the essential role of OT-expressing neurons in neural development but also expands our understanding of their extensive contributions to brain function and metabolic regulation. It identifies potential critical timing and targets for therapeutic interventions in neurodevelopmental disorders such as PWS.

## Materials and methods

### Ethics statement

All experiments were performed in accordance with the guidelines for animal use specified by the European Union Council Directive of September 22, 2010 (2010/63/EU) and the approved protocol (APAFIS# 13387–2017122712209790 for the studies performed in Lille and authorization APAFIS#27108-2020090909383268 v2 for the studies conducted in Bordeaux) by the Ethical Committee of the French Ministry of Education and Research.

### Animals

All mice were housed under specific pathogen-free conditions within a temperature-controlled environment (21–22°C), maintaining a 12-hour light/dark cycle and 40% humidity. The mice had *ad libitum* access to food and water. The strains used in this study included C57BL/6J wild-type mice (JAX #000664), Oxytocin-Ires Cre mice (JAX #024234), Rosa-CAG-LSL-HA-hM4Di-pta-mCitrine mice (JAX #026219, also called R26-LSL-hM4Di-DREADD mice), BoNT/B[loxP-STOP-loxP] mice (JAX #018056), mT/mG (JAX #007576, also called ROSAmT/mG mice), *Nos1* KO mice (JAX #002986), and *Magel2* KO (*Magel2*[tm1.1Mus], provided by Dr. Francoise Muscatelli, INMED, Marseille, France). All mice were backcrossed in a C57BL/6J background. Our study examined male and female animals. Both sexes were pooled when no sex differences were found.

### Chemogenetic inhibition of OT neurons

*Ot*-Cre mice were mated to R26-hM4Di-DREADD mice to generate mice that expressed inhibitory DREADD in OT neurons. *Ot*-Cre::R26-hM4Di-DREADD mice and R26-hM4Di-DREADD (control) mice received an intraperitoneal injection of the DREADD agonist ligand, compound-21 (C21; Cat. No. SML2392, Sigma-Aldrich, 2.5 mg/kg body weight) twice a day either from P0 to P7, or from P25 to P32, or from P63 to P70.

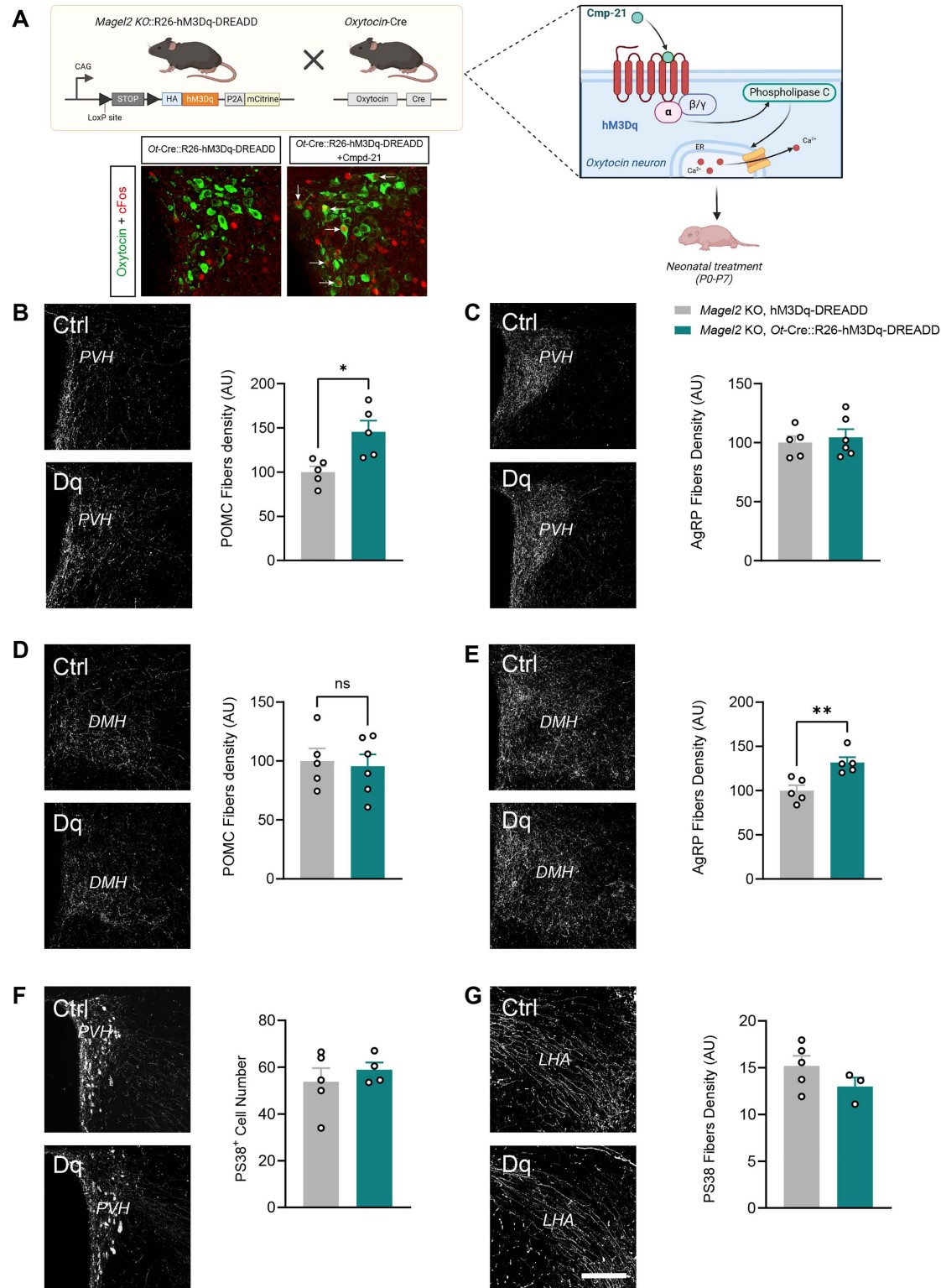

**Fig 9. Neonatal chemogenetic activation of OT neurons rescues POMC circuit deficits in *Magel2* KO mice. (A)** Description of the animal model and its validation through specific activation of OT neurons (cFos immunoreactivity) following Compound-21 injection. *Created in BioRender. Bouret, S. (2025)* https://BioRender.com/bc3g0nq. **(B–E)** Representative images and quantification of **(B, D)** POMC and **(C, E)** AgRP fibers innervating the **(B,**

C) paraventricular nucleus (PVH) and **(D, E)** dorsomedial nucleus (DMH) of adult *Magel2* KO, R26-hM3Dq-DREADD (control) and *Magel2* KO, *Ot*-Cre::R26-hM3Dq-DREADD mice injected with Compound-21 neonatally (*n* = 5−6 animals per group). **(F, G)** Representative images and quantification of OT (PS38+) neurons and fibers innervating the **(F)** PVH and (G) lateral hypothalamic area (LHA), respectively, in adult *Magel2* KO, R26-hM3Dq-DREADD (control) and *Magel2* KO, *Ot*-Cre::R26-hM3Dq-DREADD mice injected with Compound-21 neonatally (*n* = 5−3 animals per group). Data are presented as mean + SEM. *$P < 0.05$. Scale bar, 200 μm. The data underlying this Figure can be found in S1 Data.

### Inhibition of SNARE-dependent vesicular secretion in OT neurons

*Ot*-Cre mice were mated to BoNT/B$^{loxP-STOP-loxP}$ mice to generate mice expressing the botulinum toxin serotype B light chain in OT neurons, blocking exocytosis. BoNT/B were used as controls.

### Cell membrane labeling of OT neurons

To selectively label the membranes of OT neurons, *Ot*-Cre mice were crossed with mT/mG reporter mice. In this model, Cre-mediated recombination drives expression of membrane-targeted EGFP (mGFP) in OT-expressing neurons, replacing the constitutive red fluorescence (tdTomato) present in non-recombined cells. *Ot*-Cre::mT/mG mice were sacrificed at three postnatal stages: P0, P25, and P60. For P25 and P60 animals, mice were transcardially perfused with 4% paraformaldehyde, whereas brains from P0 pups were fixed by immersion in 4% paraformaldehyde for 24 hours. Brains were then processed for immunohistochemical co-labeling of GFP and PS38 as described below.

### Pharmacological blockade of OTR

Mice were injected intraperitoneally twice daily with the OTR antagonist L-368,899 (10 mg/kg BW, Cat #2641, Tocris, UK) from P0 to P7. Controls received injections with an equivalent volume of vehicle (0.9% NaCl).

### Chemogenetic activation of OT neurons in *Magel2* KO mice

*Ot*-Cre mice were mated with R26-hM3Dq-DREADD mice to generate mice that expressed stimulatory DREADD in OT neurons. These mice were then crossed with *Magel2*$^{tm1.1Mus}$ KO mice to generate *Magel2* KO, *Ot*-Cre::R26-hM3Dq-DREADD mice. *Magel2* KO, *Ot*-Cre::R26-hM3Dq-DREADD mice and *Magel2* KO, R26-hM3Dq-DREADD (control) mice received an intraperitoneal injection of the DREADD agonist ligand, compound-21 (C21; Cat. No. SML2392, Sigma-Aldrich, 2.5 mg/kg body weight) twice a day from P0 to P7.

### Immunohistochemistry

Mice were perfused transcardially at P100 with 4% paraformaldehyde for AgRP, POMC, and PS38 immunolabelings and at P10 with a mixture of 2% paraformaldehyde and 0.2% picric acid for NOS1 and PS38 co-immunolabeling. Brains were then frozen, sectioned, and processed for immunofluorescence using standard procedures [44]. The primary antibodies used for IHC were as follows: rabbit anti-POMC (1:10,000, Ref# H-029-30, Phoenix Pharmaceuticals), rabbit anti-AgRP (1:1,000, Ref# H-003-53, Phoenix Pharmaceuticals), mouse anti-neurophysin 1 clone PS38 (1:1,000, Ref# MABN844, Merck Millipore), goat anti-NOS1 (1:4,000, gift from Dr. P. C. Emson, Babraham Institute, Cambridge, UK Cat# K205, RRID:AB_2895154), goat anti-GFP (1:1,000, Ref# A6455, Thermo Fisher Scientific), and rabbit anti-cFos (1:1,000, Re# AB190289, Abcam). The primary antibodies were visualized with goat anti-rabbit IgG conjugated with Alexa Fluor 568 (1:500, Ref# A11011, Invitrogen) or goat anti-mouse IgG conjugated with Alexa Fluor 647 secondary antibodies (1:500, Ref# A21235, Life Technologies) or donkey anti-mouse IgG conjugated with TRITC (1;1,000, Cat# 715-025-151, Jackson Immunoresearch) or donkey anti-goat IgG conjugated with Alexa Fluor 488 (1:1,000, Cat# 705-545-147, Jackson Immunoresearch). Sections were mounted on Glycerol-based medium with DAPI to visualize cell nuclei.

Images were acquired using an AxioImager Z2 Apotome microscope equipped with a 40× objective (AxioCam MRm camera, Zeiss) for the quantitative analysis of POMC, AgRP, and PS38 immunolabelings and using a Leica SP8 confocal microscope equipped with a 20× and 40× objectives for the analysis of PS38+Ot-mGFP and NOS1+PS38 co-immunolabeling, respectively. Quantifications were performed in two sections per animal. Slides were numerically coded to obscure the experimental group. The image analysis was performed using the Fiji software (NIH) as previously described [44]. For the quantitative analysis of fiber density (for POMC, AgRP, and PS38), a maximum intensity projection was performed on 5 µm of the Z-stack. The threshold was set manually to ensure that only a positive signal was measured. Images were then binarized, and a standardized region of interest (ROI) was placed within the nucleus of interest. The software then calculated the number of pixels in the ROI corresponding to the signal of interest. This pixel count was finally normalized to the dimensions of the ROI, ensuring comparability across all images. The integrated intensity, which reflects the total number of pixels in the binarized image, was then calculated within the ROI of each image of the stack [4,44,45]. For the quantitative analysis of cell numbers, PS38-immunopositive cells were manually counted using the Fiji software. Only cells with corresponding DAPI-stained nuclei were included in our counts. In addition, the Imaris software was used to isolate the 3D volumes corresponding to the immunohistological labeling of PS38+ neurons and AgRP+ and POMC+ fibers in the PVH. The "shortest distance to surface" formula included in the Imaris software was then used to evaluate the proximity of the AgRP or POMC signals to the PS38 signals. This distance was used as a proxy to determine the distance between PS38+ neurons and AgRP+ or POMC+ projections.

## RNAscope fluorescent in situ hybridization (FISH)

FISH was performed on fresh-frozen brain sections derived from WT mice at P0, P5, P10, P25, and P60 using the RNAscope Multiplex Fluorescent Kit v2 according to the manufacturer's instruction (Advanced Cell Diagnosis, CA, USA) and as previously described [44]. Adjustment of protease incubation time was made for the labeling of perinatal tissues. Commercially available probes were used to detect *Otr* (ACD, Ref: 412171), *Pomc* (ACD, Ref: 314081-C2), and *Agrp* (ACD, Ref: 400711-C3) mRNAs. Sections were mounted on a glycerol-based medium with DAPI to visualize the border of each nucleus. Images were acquired using an AxioImager Z2 Apotome microscope (AxioCam MRm camera, Zeiss) and Zen 3.1 (blue edition) software. The quantitative analysis of *Otr* mRNA was performed in the ARH, VMH, DMH, and PVH by counting the number of *Otr*-positive spots within each nucleus using the Fiji software. The total number of transcripts was then normalized to the area of each respective nucleus. The quantitative analysis of *Pomc* and *Agrp* cells expressing *Otr* was categorized based on the degree of co-localization with *Otr*-positive spots. This classification system established five distinct classes of *Otr* mRNA expression: none (0 spots/cell), low (1–3 spots/cell), medium (4–6 spots/cell), high (7–9 spots/cell), and very high (10+ spots/cell).

## Isolated ARH explant culture

Brains were collected from P4 WT mice and sectioned at 200-µm thickness with a vibroslicer as previously described [46]. The ARH was then carefully dissected out of each section under a stereomicroscope. Explants were cultured onto a rat tail collagen matrix (BD Bioscience), and each explant was incubated with fresh modified Basal Medium Eagle (Invitrogen) containing OT (100 ng/ml) or vehicle (NaCl). After 48 hours, the explants were fixed in paraformaldehyde, and neurites extending from the explants were stained with a TUJ1 (βIII tubulin) antibody (rabbit, 1:5,000, Covance, cat# MMS-435P) as described previously [46]. Image analysis was performed using a Zeiss LSM 710 confocal system equipped with a 10× objective. Slides were numerically coded to obscure the treatment group. The image analysis was performed using the Fiji software (NIH) as previously described [46].

## Metabolic phenotyping

Animals were weighed using an analytical balance at P100. Body composition analysis was conducted using a Minispec LF50 Series (Bruker Corporation, Massachusetts) at P100. Fat mass, lean mass, and free body fluids measurements were expressed as a percentage of total body weight. Food intake, $O_2$ consumption and $CO_2$ production, EE, respiratory

exchange ratio (*i.e.*, $VCO_2/O_2$), locomotor activity (X and Y axis), and Z rearing were monitored in fed mice at P100 using a combined indirect calorimetry system (TSE Pheno Master Systems GmBH, Germany). The mice were acclimated in monitoring chambers for two days, and the data were collected for six days. These physiological measures were performed at the mouse metabolic phenotyping platform of the University of Lille. Glucose tolerance tests were conducted in mice at P100 through i.p. injection of glucose (2g/kg body weight) after an overnight fasting. Blood glucose levels were measured at 0, 15, 30, 60, 90, 120 min post-injection, as previously described [44].

## Statistical analysis

Values are represented as the mean + SEM or mean ± SEM. Statistical analyses were conducted using GraphPad Prism (version 10.2.2). In accordance with standard recommendations, data sets with small sample sizes ($n < 8$) were analyzed using non-parametric Mann–Whitney $U$ test. For groups ≥8, data normality was first assessed using the Shapiro–Wilk test. When the assumption of normality was violated ($p < 0.05$), appropriate non-parametric tests were applied. Dunn's test for non-parametric comparisons was applied. Statistically significant outliers were calculated using Grubb's test for outliers. $P \leq 0.05$ was considered statistically significant.

## Supporting information

**S1 Data. Metadata.**
(XLSX)

**S1 Fig. Manipulation of oxytocin neurons does not affect melanocortin circuits in female mice.** Quantification of the density of **(A, B, G, H, M, N, S, T)** POMC and **(C, D, I, J, O, P, U, V)** AgRP fibers in the **(A, C, G, I, M, O, S, U)** PVH, and **(B, D, H, J, N, P, T, V)** DMH and quantification of **(E, K, P, W)** the number of OT neurons in the PVH and **(F, L, R, X)** the density of OT fibers in the LHA of adult control and **(A–F)** *Ot*-Cre::R26-LSL-hM4Di-DREADD female mice injected with Compound-21 neonatally, or during juvenile life, or during adulthood ($n = 5$–7 animals per group), **(G–L)** *nNOS* KO female mice ($n = 5$ animals per group), **(M–R)** *Ot*-Cre::*BoNT/B* adult female mice ($n = 5$–3 animals per group), or **(S–X)** female mice injected with the OTR antagonist L-368,899 neonatally ($n = 7$–8 animals per group). Data are presented as means + SEM. The data underlying this Figure can be found in S1 Data
(JPG)

**S2 Fig. No sex difference in OT projections or *Otr* expression in arcuate neurons. (A)** Representative images of OT fibers innervating the arcuate nucleus (ARH) of P0, P25, and P60 male and female mice as revealed by an immunostaining against oxytocin (PS38 antibody) and genetic labeling of OT fibers (Ot mGFP mice). **(B)** Quantification of *Otr co-expression in Pomc and Agrp neurons* of P0, P25, and P60 male and female mice ($n = 3$–4 animals per group). Scale bar, 200 µm. The data underlying this Figure can be found in S1 Data
(JPG)

**S3 Fig. Neonatal blockade of oxytocin signaling does not affect metabolic regulations in female mice. (A)** Body weight, **(B)** body composition, and **(C)** fasting glycemia, in adult female mice injected with saline (control) or an oxytocin receptor antagonist (L-368,899, OTRA) neonatally ($n = 4$–8 animals per group). Data are presented as means ± SEM. **$P < 0.01$. The data underlying this Figure can be found in S1 Data
(JPG)

## Acknowledgments

We thank Dr. Sophie Croizier (Unversity of Lausanne) for discussions and comments on the manuscript and Antria Antreou for her assistance with the immunostaining shown in Fig 5A. We thank Meryem Tardivel and Antonino Bongiovanni

from the BICEL Photonic Microscopy. We also thank the Bordeaux Imaging Center, a service unit of the CNRS-INSERM and Bordeaux University, a member of the national infrastructure France BioImaging supported by the French National Research Agency (ANR-10-INBS-04). The technical assistance of Sébastien Marais, Magali Mondin, and Christel Poujols is acknowledged. We thank Julien Devassine (animal core facility, Lille) and members of the PLBS UAR2014-US41 for their expert technical support.

## Author contributions

**Conceptualization:** Pierre-Yves Barelle, Philippe Ciofi, Françoise Muscatelli, Sebastien G. Bouret.

**Data curation:** Pierre-Yves Barelle, Fabienne Schaller, Soyoung Park, Emilie Caron, Jessica Klucznik, Sebastien G. Bouret.

**Formal analysis:** Pierre-Yves Barelle, Françoise Muscatelli, Sebastien G. Bouret.

**Funding acquisition:** Françoise Muscatelli, Sebastien G. Bouret.

**Investigation:** Pierre-Yves Barelle, Philippe Ciofi, Françoise Muscatelli, Sebastien G. Bouret.

**Methodology:** Pierre-Yves Barelle, Fabienne Schaller, Soyoung Park, Emilie Caron, Jessica Klucznik, Sebastien G. Bouret.

**Project administration:** Sebastien G. Bouret.

**Supervision:** Françoise Muscatelli.

**Validation:** Sebastien G. Bouret.

**Writing – original draft:** Pierre-Yves Barelle, Françoise Muscatelli, Sebastien G. Bouret.

**Writing – review & editing:** Philippe Ciofi, Françoise Muscatelli, Sebastien G. Bouret.

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
