## [Editor Report · Decision Letter 0]

1 Apr 2025

Dear Dr Bouret,

Thank you for submitting your manuscript entitled "Vesicle-mediated oxytocin release drives melanocortin circuit maturation during a neonatal critical period" for consideration as a Research Article by PLOS Biology.

Your manuscript has now been evaluated by the PLOS Biology editorial staff, and I am writing to let you know that we would like to send your submission out for external peer review.

Once your full submission is complete, your paper will undergo a series of checks in preparation for peer review. After your manuscript has passed the checks it will be sent out for review. To provide the metadata for your submission, please Login to Editorial Manager (https://www.editorialmanager.com/pbiology) within two working days, i.e. by Apr 03 2025 11:59PM.

Kind regards,

Taylor

Taylor Hart, PhD,

Associate Editor

PLOS Biology

thart@plos.org

---

## [Decision Letter · Decision Letter 1]

12 May 2025

Dear Dr Bouret,

Thank you for your patience while your manuscript "Vesicle-mediated oxytocin release drives melanocortin circuit maturation during a neonatal critical period" was peer-reviewed at PLOS Biology. It has now been evaluated by the PLOS Biology editors, an Academic Editor with relevant expertise, and by several independent reviewers.

In light of the reviews, which you will find at the end of this email, we would like to invite you to revise the work to thoroughly address the reviewers' reports.

As you will see below, the reviewers consistently praise the level of interest in the findings. However, they raised a number of concerns, especially the inconsistent effects of the oxytocin manipulations on AgRP neurons, the lack of sufficient information about the developmental timeline of OT neurons in the ARH, and whether the images shown are sufficiently detailed and representative. They also noted some missing controls and areas requiring additional discussion, especially the sex differences. We think that these concerns will need to be thoroughly addressed before we can consider your study further for publication. Additionally, please note that PLOS Biology does not allow references to unpublished data and we ask that you include all relevant data needed to support the conclusions of this study.

Given the extent of revision needed, we cannot make a decision about publication until we have seen the revised manuscript and your response to the reviewers' comments. Your revised manuscript is likely to be sent for further evaluation by all or a subset of the reviewers.

**IMPORTANT - SUBMITTING YOUR REVISION**

*Re-submission Checklist*

*Published Peer Review*

*PLOS Data Policy*

*Blot and Gel Data Policy*

Sincerely,

Taylor

Taylor Hart, PhD,

Associate Editor

PLOS Biology

thart@plos.org

REVIEWS:

Reviewer #1: This manuscript presents compelling evidence that oxytocin (OT) neurons are critical for the development of melanocortin circuits in the hypothalamus, particularly through their paracrine action during a critical neonatal period. Using a combination of chemogenetic manipulation, pharmacological interventions, genetic models, and ex vivo assays, the authors demonstrate that OT influences POMC and AgRP neuronal projections to the paraventricular nucleus (PVH), with downstream implications for metabolism. They also show that early-life activation of OT neurons can rescue POMC defects in a Prader-Willi Syndrome (PWS) mouse model. This is a novel and important study with translational relevance.

Strengths

The study identifies OT as a neurotrophic factor guiding melanocortin circuit formation—a previously underexplored developmental role. The timing specificity (neonatal critical period) and sex-specific effects (largely male) are biologically significant and well-demonstrated. The translational application to PWS model rescue experiments adds substantial clinical relevance. Utilization of chemogenetic models (hM4Di/hM3Dq DREADDs) is state-of-the-art and appropriate. BoNT-B conditional mutants provide elegant mechanistic insight into the requirement of SNARE-mediated exocytosis. Ex vivo ARH explant culture adds an independent and complementary technique validating in vivo findings. Anatomical, molecular (RNAscope), physiological (metabolic phenotyping), and pharmacological tools were all applied effectively. Clear and robust figures support the conclusions, with adequate statistical handling.

Points Requiring Clarification

1) Mechanism of OT Action Remains Vague

Although OT is shown to act in a paracrine manner and OTRs are present in ARH neurons, the site of OT release remains speculative. The lack of a loss-of-function model specific to the OT peptide (as opposed to neuron inhibition or SNARE-mediated exocytosis) makes it difficult to attribute the phenotype solely to OT rather than other co-released factors. Discussion should be expanded to include these limitations of the study.

2) AgRP Circuit Inconsistencies

OT manipulations affect both POMC and AgRP neurons in some experiments but not consistently across all conditions (e.g., OTRA affects only POMC fibers). This partial effect requires mechanistic discussion, particularly since AgRP neurons express OTRs (though to a lesser extent). While the study is already robust, demonstrating the localization of OT fiber density in the ARH during early postnatal life would provide valuable information to validate the hypothesis. In the discussion, it is noted that the ARH is virtually devoid of OT fibers (Barelle & Bouret, unpublished observation). Were these observations made in young animals? It is possible that OT innervation is stronger during early development, which could account for the neurotrophic effects of OT.

3) Sex Differences Underexplored

The striking male-specific effects are presented but not mechanistically addressed. The authors could speculate whether hormonal differences or OT/OTR expression levels contribute.

4)Minor Issues by Figure

Figure 5

Panel 5C indicates ARC instead of ARH

Figure 7, Figure 8 and Suppl Fig 1

The figure legend refers to "L-368,899," but the compound is called "L-368,889" in the rest of the manuscript. This is likely a typo and should be corrected for consistency.

This is an elegantly designed, technically robust, and conceptually impactful study that addresses an important gap in neurodevelopmental and metabolic biology. I recommend publication pending minor revisions.

Reviewer #2: This study by Barelle et al. presents compelling evidence for a novel neurodevelopmental role of oxytocin (OT) neurons in establishing hypothalamic melanocortin system connections. Using a robust, multi-modal approach including chemogenetics, SNARE-blockade, receptor antagonism, explant assays, and a genetic rescue in a Prader-Willi syndrome model, the authors demonstrate that OT secretion during a specific neonatal period is critical for guiding POMC and AgRP axon projections to the PVH, impacting long-term metabolic outcomes. The research is notable for its comprehensive methodology, clear identification of a critical temporal window, linkage of anatomical changes to physiological function, and its translational relevance, making the findings novel and of broad interest.

However, several points require clarification to strengthen the conclusions:

1. Could the authors provide more detail on the developmental timeline of POMC and AgRP axon innervation into the PVH relative to the appearance of OT neurons/fibers there (perhaps referencing Figure 2 or adding supplemental information)? This context is crucial to determine whether the neonatal chemogenetic inhibition of OT neurons affects the initial targeting and innervation by POMC/AgRP axons or the subsequent maintenance of these connections.

2. The study does not definitively establish a direct, cell-specific role for OT signaling on the target neurons (POMC/AgRP). While the current data are suggestive, utilizing conditional OTR knockout mice would be required to unequivocally demonstrate that OT acts directly on these populations to guide their projections. This limitation should be acknowledged in the Discussion.

3. The possibility that OT neurons might co-release other relevant axon guidance molecules during this developmental window, which could contribute to the observed effects independently of, or in addition to, OT peptide signaling, has not been ruled out. This alternative possibility should also be considered in the Discussion.

4. The observed sex differences, particularly the noted lack of effect in females for some manipulations, are intriguing but currently under-explored in the Discussion.

Several minor points could also improve the manuscript's clarity and rigor:

* Figure Legends: While comprehensive, the figure legends could potentially be condensed for better readability, perhaps by moving extensive methodological details to the main Methods section.

* Methods Clarity: The Methods section should specify the concentration of OT used in the explant cultures and include justification for its physiological relevance.

* Statistical Reporting: Regarding statistics, please clarify how normality was assessed prior to choosing non-parametric tests and confirm whether appropriate multiple-comparison corrections were applied where necessary.

Reviewer #3: This study described a series of complementary experiments on a neurotropic action of oxytocin during early developmental periods on the melanocortin POMC and AgRP projections to PVH. The experiments appear to be carefully designed, executed and appropriately analyzed. The results on the rather important role of oxytocin are exciting and will be interesting to the field. However, several important issues were noted that need to be addressed.

1) Pictures in Fig. 2C-2D, Fig.3B-3C and Fig.7B do not seem consistent with the conclusion shown in the statistical analysis. These pictures are suggested to be replaced to avoid unnecessary confusion. Importantly, given the importance of the validity of this comparison, it is suggested that the authors show a series of PVH sections to show a convincing comparison between the two conditions.

2) There seems to have some inconsistency between neuron activity changes and oxtyocin receptoer activation on AgRP projections. This inconsistency needs to be clarified.

3) Fig. 9 demonstrated the effect on Magel2KO phenotypes. The data on the KO as well as Gq control should be provided.

Reviewer #4: Barelle et al. provide compelling evidence linking the oxytocin system to the development of melanocortin circuitry. The study is timely and interesting and will be of interest to many working in energy homeostasis and oxytocin-related research. Overall, the paper is well written. However, I have some additional questions outlined below. I believe most of these can be addressed with additional discussion, although a few more control experiments are likely warranted.

-Did the authors validate that Cmpd-21 reduced activity of oxytocin neurons in hM4Di expressing mice? It is quite difficult to chronically inhibit neurons with DREADDs and some evidence should be provided to show that the DREADD inhibition worked. I don't think this is a major concern given that a similar phenotype was shown with the iBot experiment (a different form of inhibition) but this data should be provided if possible.

-Likewise is there any validation of the chemogenetic activation (i.e. increased fos in Oxt neurons)?

-Do the authors have any potential explanation for the male specific effects of oxt neuron inhibition? Some discussion of this in the discussion section seems warranted

-Do the authors have any explanation for the lack of effect on AgRP projections with the antagonist, but not with the chemogenetic bottox approaches?

-The metabolic phenotype resulting from Oxt inhibition is hard to understand. How do the authors explain the lack of a change in body weight given increased feeding and no change in energy expenditure? It is also perplexing that a phenotype that impairs melanocortin pathways leads to measures suggesting improved metabolism (i.e. lower adiposity, elevated lean mass, improved glucose tolerance). What is the physiological relevance of this pathway for circuit development given that disrupting it improves metabolism?

-What is the effect of OT neuron stimulation in WT mice? Is the increase in POMC specific to the Magel2 KO model?

-How does the explant experiment relate to AgRP/POMC projections? I assume the fiber growth in this model is not specific to just these neurons. Does this suggest that Oxt may regulate many other arcuate projections to PVN, and if so do you see changes in other arcuate projections to PVN following blockade of Oxt? Perhaps this may help to explain the complicated metabolic phenotype?

-Is representative image in 4E for iBot representative? It looks like the POMC signal is much lower in iBot group.

---

## [Editor Report · Decision Letter 2]

13 Oct 2025

Dear Dr Bouret,

Thank you for your patience while we considered your revised manuscript "Vesicle-mediated oxytocin release drives melanocortin circuit maturation during a neonatal critical period" for publication as a Research Article at PLOS Biology. This revised version of your manuscript has been evaluated by the PLOS Biology editors and the Academic Editor.

Based on our Academic Editor's assessment of your revision, we are likely to accept this manuscript for publication. Please also make sure to address the following data and other policy-related requests.

IMPORTANT: Please ensure that your next revision addresses the following editorial requirements:

---------------

**Title:

-- We suggest slight tweak of your title to better align it with the main findings:

"Oxytocin neurons drive melanocortin circuit maturation via vesicle release during a neonatal critical period"

**Financial disclosure statement:

-- Please add links to the funding agencies in the Financial Disclosure statement in the manuscript details.

**Competing interests:

-- As Dr Bouret is a member of the PLOS Biology editorial board, please change your Competing Interests to the following statement:

“SGB is a member of PLOS Biology’s Editorial Board. The other authors declare that no competing interests exist."

**Data:

-- Please cite the location of the data clearly in all relevant main and supplementary Figure legends, e.g. “The data underlying this Figure can be found in S1 Data” or “The data underlying this Figure can be found in https://doi.org/10.5281/zenodo.XXXXX”

-- Thank you for providing numerical data in your file titled "Barelle - Metadata". Can you please change the name of this file to "S1 Data" and ensure that it is are invariably referred to (in the manuscript, figure legends, and the Description field when uploading your files) this way?

**Abstract

-- Please note that per journal policy, the model system/species studied should be clearly stated in the abstract of your manuscript.

---------------

We expect to receive your revised manuscript within two weeks.

*Published Peer Review History*

*Press*

Sincerely,

Taylor

Taylor Hart, PhD,

Associate Editor

thart@plos.org

PLOS Biology

---

## [Editor Report · Decision Letter 3]

28 Oct 2025

Dear Dr Bouret,

Thank you for the submission of your revised Research Article "Oxytocin neurons drive melanocortin circuit maturation via vesicle release during a neonatal critical period" for publication in PLOS Biology. On behalf of my colleagues and the Academic Editor, Yi-Ping Hsueh, I am pleased to say that we can in principle accept your manuscript for publication, provided you address any remaining formatting and reporting issues. These will be detailed in an email you should receive within 2-3 business days from our colleagues in the journal operations team; no action is required from you until then. Please note that we will not be able to formally accept your manuscript and schedule it for publication until you have completed any requested changes.

PRESS

Sincerely, 

Taylor

Taylor Hart, PhD,

Associate Editor

PLOS Biology

thart@plos.org